# State-of the-Art Constraint-Based Modeling of Microbial Metabolism: From Basics to Context-Specific Models with a Focus on Methanotrophs

**DOI:** 10.3390/microorganisms11122987

**Published:** 2023-12-14

**Authors:** Mikhail A. Kulyashov, Semyon K. Kolmykov, Tamara M. Khlebodarova, Ilya R. Akberdin

**Affiliations:** 1Department of Computational Biology, Scientific Center for Information Technologies and Artificial Intelligence, Sirius University of Science and Technology, 354340 Sochi, Russia; kulyashov.ma@talantiuspeh.ru (M.A.K.); kolmykov.sk@talantiuspeh.ru (S.K.K.); tamara@bionet.nsc.ru (T.M.K.); 2Department of Natural Sciences, Novosibirsk State University, 630090 Novosibirsk, Russia; 3Department of Systems Biology, Institute of Cytology and Genetics SB RAS, 630090 Novosibirsk, Russia; 4Kurchatov Genomics Center, Institute of Cytology and Genetics SB RAS, 630090 Novosibirsk, Russia

**Keywords:** genome-scale metabolic modeling, constraint-based modeling, context-specific modeling, pipeline, tool, transcriptomics, methanotrophy

## Abstract

Methanotrophy is the ability of an organism to capture and utilize the greenhouse gas, methane, as a source of energy-rich carbon. Over the years, significant progress has been made in understanding of mechanisms for methane utilization, mostly in bacterial systems, including the key metabolic pathways, regulation and the impact of various factors (iron, copper, calcium, lanthanum, and tungsten) on cell growth and methane bioconversion. The implementation of -omics approaches provided vast amount of heterogeneous data that require the adaptation or development of computational tools for a system-wide interrogative analysis of methanotrophy. The genome-scale mathematical modeling of its metabolism has been envisioned as one of the most productive strategies for the integration of muti-scale data to better understand methane metabolism and enable its biotechnological implementation. Herein, we provide an overview of various computational strategies implemented for methanotrophic systems. We highlight functional capabilities as well as limitations of the most popular web resources for the reconstruction, modification and optimization of the genome-scale metabolic models for methane-utilizing bacteria.

## 1. Introduction

Biosystems engineering, or synthetic biology, is one of the rapidly developing fields in modern biology which include an array of different approaches to design novel and complex biological systems with specified properties often on the basis of in-depth genome reorganization or even its de novo synthesis. However, its development requires fundamental knowledge of the structural and functional organization of genomes, transcriptomes, proteomes, and metabolomes. Thus, it is not surprising that synthetic biology is deeply rooted in systems biology approaches, which provide a holistic and quantitative understanding of existing biological systems and validated for a narrow set of model microbial systems, such as *E.coli* or *Bacillus* [1,2,3,4,5,6,7,8]. The implementation of the systems biology potential toward non-model systems often starts with the genome-scale metabolic (GSM) modeling following by its validation using omics data or detailed analyses of gene interactions via reconstruction of regulatory networks [9]. Here we summarize the systems biology approaches that have been applied toward understanding methane metabolism, known as methanotrophy.

Methanotrophy, as a metabolic capability to convert methane, has drawn the attention of systems biologists as a potential platform for capturing methane and conversion of it into novel fuels or chemicals to solve the challenge associated with greenhouse gases driven climate change or pollution [10,11,12]. The accumulation of high-throughput data as transcriptomics and proteomics for methanotrophs provides an opportunity to specify bacterial metabolism by adjusting the fluxes bounds for each metabolic reaction via observed expression levels and gene-protein reaction (GPR) rules on which the majority of algorithms for the reconstruction of context-specific GSM models are relied on [13,14]. This review compiles the current GSM models of methanotrophic bacteria and describes how they have been applied to investigate the methane metabolism in a holistic fashion. We also highlight gaps in the development and analysis of context-specific metabolic models for methanotrophs and summarize computational tools and web-resources that can streamline the constraint-based modeling of microbial metabolism from the reconstruction of GSM models to the development of context-specific metabolic models considering transcriptomic datasets to fine-tune the flux bounds of an original model to a specific experimental context.

## 2. Reconstruction and Analysis of Genome-Scale Metabolic Models

### 2.1. The Stages of Metabolic Model Reconstruction

The development of a GSM model of any metabolic process involves several fundamental steps (Figure 1). One of the first steps is the reconstruction of the metabolic network, which is performed based on the annotation data of the sequenced genome of an organism of interest and includes information about the genes and proteins/enzymes encoded by them, biochemical reactions of the analyzed metabolic pathway and metabolites [15,16]. The sources of this information are databases and web-portals which will be discussed below.

The functionality of this network is confirmed at the next stage of model reconstruction by additional information from published data and experiments conducted for model organisms and/or closely related species [17,18].

The third step adds species-specific physiological, biochemical, physical and phenotypic characteristics of the network components including the thermodynamic and kinetic parameters of reactions and the metabolites available in publications or databases.

The resulting metabolic map allows one to mathematically link enzymatic reactions and the metabolites participating in them as substrates or products in a certain stoichiometry using a stoichiometric S matrix.

Generally, for GSM models validation it is assumed that the biological system is in a quasi-equilibrium state, i.e., the concentrations of metabolites do not change in the system, and therefore the right-hand sides of the system of differential equations describing the change in the concentration of metabolites can be equated to zero. Thus, a system of linear algebraic equations is obtained. In other words, the assumption that the metabolic system has reached a quasi-equilibrium implies that the sum of all reaction fluxes in which a certain metabolite is synthesized is equal to the sum of enzymatic reaction fluxes in which this metabolite is consumed.

These constraints on flux balances are mathematically formulated as S∗V=0, where *S* is the stoichiometric matrix and *V* is the vector of reaction rates (fluxes) of the studied metabolic system. As a result, this mathematical expression was performed under quasi-equilibrium conditions and additionally introduced constraints on the rates of intracellular reversible and irreversible reactions (on the lower (LB) and upper boundaries (UB), respectively), as well as on the reactions of transport exchange between the compartments of the model, which enables one to conduct a flux balance analysis (FBA) using linear programming methods at the last stage of model development. FBA addresses one of the optimization problems [15,16,19]. For example, optimizing the production of cellular biomass or one of the targeted, biotechnologically important products under wild-type phenotype conditions and/or under various genetic modifications (knockouts, increased expression of a gene encoding a particular enzyme) [20,21,22].

A GSM model constructed in this way requires further refinement based on available experimental data for a more adequate description of the metabolism of the object under study. It will ultimately provide more relevant and accurate predictions of phenotypic changes in the growth of the bacterium under certain conditions of the culturing or as a result of genetic modifications employing in silico experiments [20,23,24,25,26].

### 2.2. Databases of the Microorganisms’ Genomes

Numerous databases and web-portals resources have been developed and are available for the initial metabolic pathway reconstruction, including BioCyc [27], KEGG [28], GenBank [29], Ensembl Bacteria [30], PATRIC [31], MicroScope [32] and IMG/M [33]. Below we briefly summarize each resource.

**BioCyc** (https://biocyc.org/) is a web-portal of prokaryote genomes that integrates sequenced genomes with expert-processed an information from published data as well as an information imported from other biological databases. The BioCyc collection consists of over 20,040 pathway/genome databases (PGDBs) [27], each containing the complete genome and putative metabolic network of a single organism, which is predicted by the Pathway Tools software and comprises metabolites, enzymatic reactions and metabolic pathways [34]. BioCyc provides extensive search and visualization tools, as well as toolkits for omics data analysis, comparative genomic analysis, metabolic pathways search, and metabolic model generation. BioCyc expert analytical information includes experimental data on gene functions, kinetic parameters of enzymatic reactions, enzyme activators and inhibitors. The database also contains textual mini-reviews authored by expert curators that summarize information on enzymes and pathways with corresponding references [27,35]. The main drawbacks of this resource now are its limited use without a paid subscription.

**KEGG** (Kyoto Encyclopedia of Genes and Genomes, https://www.genome.jp/kegg/) is manually curated resource represented by a set of databases and associated bioinformatics software for analyzing and modeling the functional behavior of a cell or higher-order organism based on information about its genome. KEGG includes both data relevant for biomedical research (e.g., KEGG DISEASE and KEGG DRUG) and tools for the analysis of bulk molecular data [28,36,37]. Of particular note are the KEGG PATHWAY metabolic maps, which is a powerful tool in the reconstruction of GSM models enabling the analysis of metabolic pathways for a selected organism.

**UniProt** (https://www.uniprot.org/, [38]), **Brenda** (https://www.brenda-enzymes.info/index.php, [39]) and **Sabio-RK** (http://sabio.h-its.org/, [40]) are also very useful and widely cited resources for biochemical data and enzymes annotation that are essential for proper metabolic pathway reconstruction.

Whereas **Genbank** (https://www.ncbi.nlm.nih.gov/genbank/) is an annotated collection of publicly available nucleotide sequences for more than 500 000 formally described species [41], **Ensembl Bacteria** (https://bacteria.ensembl.org/index.html)—a portal containing specifically bacterial and archaea genomes as well as a collection of data on genes and the proteins they encode [42]. Ensembl has BLAST and an algorithm based on hidden Markov models as a tool to seek protein motifs. Pan-taxonomic comparison tools are available for key microbial species. The current version of the portal also presents genome annotation capabilities, includes transcriptome data, and supports comparative analysis [30]. However, this resource lacks tools for reconstructing and analyzing metabolic pathways.

**PATRIC** (http://www.patricbrc.org) is designed to support biomedical research aimed at studying bacterial infectious diseases through the integration of pathogen information using available data and tools for analysis. Integrated data covers genomics, transcriptomics, protein-protein interactions, 3D protein structures and metadata from various organisms. PATRIC provides genome assembly and annotation as well as RNA-seq data analysis [31,43,44].

**MicroScope** (https://mage.genoscope.cns.fr/microscope/home/index.php/) is a web resource of prokaryotic genomic sequences, similar to IMG/M, with a large set of tools for genome annotation, comparative analysis and visualization, which is especially important for the validation of the functional annotation quality. Expert verification of annotations and genomes by microbiologists is in progress. The latest version of the database expands the set of tools for functional gene annotation by identifying orthologous genes and viral regions (virulomes) and for predicting regions associated with antibiotic resistance too [32].

**Integrated Microbial Genomes & Microbiomes** (**IMG/M**, https://img.jgi.doe.gov/)—a web-based resource for the annotation and analysis of genomic sequences integrated with different types of metadata, encompassing microbiome composition and environmental factors. The resource contains information on sequenced genomes of various species of organisms (bacteria, archaea, eukaryotes, plasmids and viruses) and provides toolkits to analyze them. In addition, IMG comprises datasets that are imported from publicly available sources such as NCBI Genbank, SRA and DOE National Microbiome Data Collaborative (NMDC) or submitted by external users [33,45]. Furthermore, several efforts are underway to streamline the integration of the IMG/M tools with other resources for mathematical modeling, e.g., Kbase [33].

### 2.3. GSM Models for C1-Utilizing Bacteria

The databases described above serve as the basis for construction of GSM models for prokaryotes, including models for unique groups of microorganisms such as methanotrophs and methylotrophs. These are bacteria and archaea that utilize C1-containing hydrocarbons as their sole carbon sources for growth. The developed models are employed to study the metabolic capabilities of diverse strains of C1-utilizing bacteria, including growth on methane and/or methanol using various metabolic pathways. These models also can be used to predict more efficient ways for the production of target value-added compounds and to study the peculiarities of their metabolism under different cultivation conditions (see reviews: [10,11,46,47]).

Table 1 summarizes the currently available information on the strains of methanotrophic bacteria for which GSM models have been developed as well as model parameters like the number of considered genes, metabolic reactions and intracellular metabolites produced. It is worth noting that several detailed reviews on built GSM models describing the metabolism of C1-utilizing bacteria and their application of biotechnological problems have been presented previously [11,46,47].

We briefly describe the issues for which these models were developed. For example, models for a well-studied organism such as *M. extorquens* AM1, which is a facultative methylotroph, were applied to study central cell metabolism and key steps in C1 assimilation. The unique topology of the core metabolic network and its metabolic fragility in *M. extorquens* were identified on the basis of the combination of genome-scale metabolic modeling and experimental approaches [48,49]. The ability to grow and switch to different multi-carbon sources was also shown for *M. extorquens* AM1 [49].

The *i*Mb5G model for *M. buryatense* 5G, the first published GSM model of methanotrophic bacteria, was used to interrogate the feasibility of three possible modes of methane oxidation (redox-arm, the direct coupling mode and uphill electron transfer) and to test the efficiency of carbon conversion via different C1 utilization pathways including variants of ribulose monophosphate pathway and the serine cycle. The extended version, *i*Mb5GB1 was applied to explore the ability of the methanotrophic strain to be a fatty acid producer [50,51].

The *i*IA407 model for a closely-related strain, *M. alcaliphilum* 20Z^R^, was constructed based on genomic, enzymatic and transcriptomic data and refined using published ^13^C-carbon-labeling [60] and original continuous cell culture parameters, which enabled the uncovering of the reversibility of the phosphoketolase reaction leading to the carbon flux from acetyl-CoA to xulylose-5-phosphate and highly branched TCA cycle [24]. Furthermore, a slightly extended version of the model, *i*IA409, was also applied to investigate the mechanisms of improved growth vs. carbon conversion for the strain growth in different media contents [52].

The *i*McBath and *i*MC535 models for *Methylococcus capsulatus* (Bath), which is an obligate methanotroph, were also built to study the pathways and mechanisms of the methane utilization in order to estimate methanotroph’s biotechnological potential and pave the way for rational strain design [26,53].

A series of GSM models was developed for several representatives of type II (alpha-proteobacteria) methanotrophs that use the serine cycle for carbon assimilation [54,55,57]. The built GSM models enabled the exploration of distinct features of the metabolism in *Methylocystis* species and *Methylocella silvestris* (redox-arm mechanisms as a general feature of type II methanotrophs, growth on C1 and C2 compounds, influence of the nitrogen source and mechanisms of the poly-3-hydroxybutyrate (PHB) accumulation). The metabolic models provide an effective in silico basis for the development of metabolic engineering platforms for these particular strains.

The *i*JV806 model describing the metabolism of *Methylomicrobium album* BG8, another representative of obligate aerobic gammaproteobacterial methanotrophs was recently reconstructed to study the metabolic states of the strain under growth on methane or methanol promoting biomass production and excretion of carbon dioxide and organic acids. The last ones can be considered valuable compounds for proposing the biotechnological potential of *M. album* BG8 [59].

To assess the quality of the GSM models developed for methanotrophs, we have collected all published SBML model versions (https://gitlab.sirius-web.org/RSF/Methanotrophs_models, accessed on 3 November 2023) for the first time as far as we know and applied state-of-the-art tool, MEMOTE [61] for the study. As can be seen from Table 1, the top five metabolic models according to Memote score are the ones built for *Methylococcus capsulatus* (*i*McBath), *Methylotuvimicrobium buryatense* 5G (*i*Mb5GB1), *Methylomicrobium album BG8* (*Methylomicrobium album BG8*), *Methylotuvimicrobium alcaliphilum* 20Z^R^ (*i*IA409) and *Methylosinus trichosporium OB3b* (*iMsOB3b*). Memote reports that all models are also available via the gitlab project. Additionally, we estimated the model consistency using Cobrapy to detect blocked reactions based on the flux variability analysis (FVA), including a priori inconsistent reactions like exchange, sink, demand and pseudo-reactions. As a result, the top five metabolic models according to quality score were the same as the previous analysis with the exception of *Methylomicrobium album BG8*. Moreover, we analyzed all models using FROG analysis (https://www.ebi.ac.uk/biomodels/curation/fbc), which also reports the effects of gene and reaction deletions in the provided medium. This recent community standard provides an estimation of the model quality by assessing the reproducibility of the model simulations presented in the original study. All results are available on above-mentioned gitlab project.

Anaerobic methanotrophic (ANME) archaea, which are significant contributors to the diminishment of the methane flux to the atmosphere, use alternative electron acceptors, such as nitrate or sulfate, to oxidize methane. As for the application of GSM models to investigate the metabolism of ANME, the first detailed metabolic model for a representative of these anaerobic methanotrophs, *Methanoperedens nitroreducens*, has been published relatively recently. Moreover, the authors provided a comparative analysis of the proposed model with previously constructed models for other archaea-methanogens [62].

### 2.4. Web Resources and Tools for Automatic Reconstruction of GSM Models 

Currently, there is a fairly large number of web resources and programs for the automatic reconstruction of GSM models (see review: [63]). Table 2 lists the most popular and advanced programs in terms of their abilities in model reconstruction, which we have divided into three groups for ease of comparison: web services, GUI programs and Packages/Command line programs. Below is their consideration in more detail.

#### 2.4.1. Web Resources

One of the most popular web-based resources for reconstructing and analyzing GSM models is Kbase [18], which not only offers automatic model reconstruction, but also provides modules for sequencing data processing. Kbase contains more than 160 applications, including the analysis of user data from raw short reads to fully assembled and annotated genomes, followed by the ability to analyze transcriptome data and develop metabolic models. The set of tools implemented in Kbase makes it possible to build a complete pipeline for the reconstruction and analysis of a GSM model. Moreover, Kbase gives an opportunity for model visualization as with the Kbase tools, which presents the model as a connected graph of reactions and metabolites. In addition, the distribution of fluxes can be visualized via Escher metabolic maps [64]. Kbase also enables users to integrate their original code into data analysis and also allows the addition of external applications.

**ModelSEED** is a web resource linked to Kbase that supports the creation of GSM models not only for microorganisms, but also for plants. It allows the use of a linked RAST profile (http://rast.nmpdr.org) [65] with user-annotated genomes or using existed annotations from the PATRIC database (http://www.patricbrc.org). Users can also choose their own FASTA annotation file. This version of the resource provides synonyms for reactions and metabolites in other databases, and supports the gap-filling algorithm, with the option to use a reaction file from the user [66]. A visualization of models via the Escher web tool has recently become available in ModelSEED according to the official website: https://modelseed.org/. The updated version of the reconstruction pipeline, ModelSEED v2 (MS2), has been released with improved representation of energy metabolism [67].

**FAME** (flux analysis and modeling environment) is also a web-based tool for the development of GSM models. It can be employed for generation, editing, running and analysis/visualization tasks in a single program [68]. The main distinguishing FAME feature is that analysis results can be visualized on the generally accepted KEGG metabolic map. But this is also its essential limitation, since models cannot be created for microorganisms that are not in KEGG. It should be noted that the web service is not available at this moment (verified on 27 July 2023).

**MicrobesFlux** is another web resource for GSM models reconstruction, which enables model building based on information about reactions and metabolites from the KEGG database, similar to FAME [69]. The source code is currently in the public domain, but the resource itself is not available.

**Table 2 microorganisms-11-02987-t002:** Tools for GSM models generation.

Program	Tool Type	Type of Reconstruction	Databases for Reaction Information	Programs Availability	Reference
Kbase http://kbase.us	Web-service	automatic	ModelSEED	Available	[18]
ModelSEED http://www.theseed.org/models/	Web-service	automatic	ModelSEED	Available	[66,67]
FAME http://f-a-m-e.org	Web-service	automatic	KEGG	Not available	[68]
Pathway tools ttp://pathwaytools.com	GUI based program		MetaCyc Template models	Available, but works with BioCyc license	[34]
GEMSiRV http://sb.nhri.org.tw/GEMSiRV	GUI based program	semi-automatic	Template models	Not available	[70]
AuReMe http://aureme.genouest.org	Command line program	automatic	MetaCyc, BiGG, ModelSEED	Available	[71]
Merlin v.4 https://www.merlin-sysbio.org/	GUI based program	semi-automatic	KEGG, BiGG	Available	[72]
Gapseq https://github.com/jotech/gapseq	Command line program, R package	automatic	MNXref, KEGG, BiGG, MetaCyc, ModelSEED	Available	[73]
AutoKEGGRec https://www.ntnu.edu/almaaslab and https://github.com/emikar/AutoKEGGRec	Matlab package	automatic	KEGG	Available but needed Matlab. Last update more than 5 years ago	[74]
RAVEN v2 https://github.com/SysBioChalmers/RAVEN	Matlab package	semi-automatic	KEGG, MetaCyc Template models	Available, but needed Matlab	[75]
MicrobesFlux http://tanglab.engineering.wustl.edu/static/MicrobesFlux.html	Web-service	automatic	KEGG	Not available	[69]
ScrumPy https://mudshark.brookes.ac.uk/ScrumPy	Python package	semi-automatic	BioCyc	Available	[76]
CarveMe github.com/cdanielmachado/carveme	Command line program, Python package	automatic	BiGG	Available, but needed commercial solvers (IBM CPLEX or Gurobi)	[77]
PADMet (AuReMe) https://pypi.python.org/pypi/padmet and https://gitlab.inria.fr/maite/padmet	Python package		MetaCyc, BiGG	Available	[71]
MetaDraft https://systemsbioinformatics.github.io/cbmpy-metadraft/	GUI based program	semi-automatic	Template models	Available	[78]
moped https://gitlab.com/marvin.vanaalst/moped-publication-2021	Python package	semi-automatic	MetaCyc, BioCyc	Available	[79]
Reconstructor http://github.com/emmamglass/reconstructor	Command line program, Python package	automatic	KEGG ModelSEED	Available	[80]
Bactabolize github.com/kelwyres/Bactabolize	Command line program, Python package	automatic	BiGG	Available	[81]
AuCoMe https://github.com/AuReMe/aucome	Command line program, Python package	automatic	MetaCyc	Free access, but needed Pathway tools	[82]

#### 2.4.2. GUI-Based Desktop Programs

One of the most popular programs for reconstructing GSM models is **Pathway Tools** (**Ptools**), which supports the construction and maintenance of databases specific to the organism under study (PGDB), and also has the ability to work in a web application, although its functionality is partially limited there [34]. Its features interactively explore, visualize and edit various components of the reconstructed model, such as genes, operons, enzymes (including transporter proteins), metabolites, reactions and metabolic pathways; analyze omics data taking into account the reconstructed metabolic map; and even develop microbial community models.

Other existing programs are also in this area of interest. For example, the **Merlin** program (https://www.merlin-sysbio.org/) has a user-friendly GUI and allows semi-automatic model reconstruction and its editing based on the KEGG database data [72]. Furthermore, the BiGG database was recently added to the tool as a source for model reconstruction [83]. Additionally, Merlin provides the opportunity to visualize the model using the Escher program, unlike other GUI-based tools. The last, but very essential advantage of the Merlin is a high-quality tutorial for working with the program.

**MetaDraft 0.9.2** is a full-featured platform with a graphical interface for genome-scale metabolic model reconstruction. It utilizes a constantly updated, user-expandable database of template models [78]. **GEMSiRV** is a GUI platform [70] that also uses templates from already existing mathematical models for model reconstruction. This program provides model editing and visualization using built-in tools.

#### 2.4.3. Packages and Command Line Programs

This group of resources includes a number of programs, which are presented below.

**ScrumPy** (https://mudshark.brookes.ac.uk/ScrumPy) is one of the first Python-based flexible packages for the reconstruction and analysis of metabolic models. A GSM model is directly constructed from the BioCyc Pathway Genome Database. Moreover, the tool has a modular model definition language that enables the tracking of changes during the model development process and the definition of metabolic subsystems separately [76].

The **AuReMe** (http://aureme.genouest.org/) program enables the reconstruction of genome-scale models based on information from the MetaCyc, BiGG and KEGG databases [71]. Its key feature is distribution through the Docker container, which eliminates compatibility issues between different program components.

**gapseq** (https://github.com/jotech/gapseq) is a tool for metabolic pathways prediction and automatical reconstruction of bacterial metabolic models using a curated reaction database and a novel gap-filling algorithm [73]. The program is written in the R language and distributed through the R package repository, Cran.

A number of programs allow the automatic reconstruction of models using MATLAB. Examples of such tools are **AutoKEGGRec** [74] and **RAVEN 2.0** [75]. The advantage of these programs is the compatibility with COBRA Toolbox 3, which gives the possibility to perform reconstruction and further analyze the model within the same project. AutoKEGGRec is a simple program for the automatic reconstruction of organisms based on KEGG data, which supports the reconstruction of models for several organisms represented in the KEGG database at once. It should be noted that this program has not been updated for more than 5 years. RAVEN v2, unlike AutoKEGGRec, allows the reconstruction of models not only on the basis of the KEGG database, but also using MetaCyc database and templates of existing models.

There are also programs written in Python such as **CarveMe** [77], **moped** [79], **Reconstructor** [80], **Bactabolize** [81] and **AuCoMe** [82] that provide the possibility of reconstructing the model using their own resources.

**CarveMe** (https://github.com/cdanielmachado/carveme) uses expert-curated GSM models from the BiGG database [83] to reconstruct models as initial templates. To improve the quality of reconstructed models, the program also has its own gapfilling algorithm based on the bottom-up approach. A limitation in working with CarveMe is the need to use commercial solvers such as MBI CPLEX and Gurobi.

**moped** (https://gitlab.com/qtb-hhu/moped) is a Python-based package which provides an opportunity for GSM model reconstruction from a genome sequence or by importing data from SBML [84] file or the MetaCyc or BioCyC databases as a PGDB flat file using the BLAST algorithm. Unlike most other tools, it uses a topological gap-filling algorithm [85], which is a crucial step at the process of GSM models reconstruction. Moreover, it includes a list of methods for FBA, topological model analysis and also moped model objects that are easily converted into Cobrapy model objects that simplify the integration with a large number of python-based tools for model simulation and analysis [79].

**Bactabolize** (https://github.com/kelwyres/Bactabolize) is a new command line program that also employs the BiGG database to reconstruct GSM models [81]. This tool was validated in the modeling a pathogenic strain of *Klebsiella pneumoniae* and demonstrated better model reconstruction compared to CarveMe. Bactabolize is distributed through the conda environment, which allows, as in the case of the Docker container, the elimination of the potential problems of program version incompatibility. By using the Cobrapy Toolkit, the reconstructed models are compatible with this package. Bactabolize also provides the ability to simulate the model in order to analyze the effect of single mutations on cell growth and to predict the substrates required for cell growth.

**Reconstructor** (https://github.com/emmamglass/reconstructoris a new Python package, that, unlike CarveMe and Bactabolize, uses KEGG and ModelSEED databases for model reconstruction [80]. This tool has the ability to gapfill existing models harnessing its own pFBA-based algorithm. In addition, Reconstructor has direct compatibility with Cobrapy Toolbox. A limitation of this package is that it cannot be used on Linux machines, since only Windows and MacOSX systems are currently supported.

**AuCoMe** (https://github.com/AuReMe/aucome) is not independent tool, but it is a workflow for the reconstruction of several models, making it possible to compare them with each other. It is based on the Pathway Tools reconstruction discussed above. This workflow is distributed by using both Docker and Singularity containers, and directly through the PyPi repository. It has a large set of tools for analyzing reconstructed models. It should be noted that this workflow is under development and is not a finished version [82].

### 2.5. Web-Resources and Tools for Analysis of GSM Models 

The most demanded resources for the analysis of GSM models are presented in Table 3. These tools are essential for calculation and visualization of fluxes distribution on metabolic networks predicted by different constraint-based methods. They also enable to conduct in silico experiments that can simplify the procedure to explore potential targets of gene manipulation at the systems level. This, in turn, may further scale the ability to synthesize a product in the microbial host-chassis methanotrophs compared to in vivo experiments’ labor- and time-consumes.

**COBRA Toolbox** (COnstraint-Based Reconstruction and Analysis) is one of the most widely used tools for handling GSM models. It includes methods of reconstruction and modeling, topological analysis, network visualization, as well as network integration of metabolic, transcriptomic, proteomic and thermodynamic data [86]. It contains a set of software available for use in the MATLAB program.

**COBRApy** is a software package for modeling represented by COBRA methods and written in the Python programming language. Inheriting the many strengths of the Python language, COBRApy provides the core capabilities of COBRA modeling and has a dedicated module for interfacing with the COBRA Toolbox [92]. COBRApy makes it possible to integrate models with databases and other data sources and does not require commercial software such as MATLAB.

**OptFlux** is an open-source software written in the Java programming language. Opflux is the first tool that to enable optimization problems aimed at identifying target genes and/or reactions for metabolic engineering using evolutionary algorithms or the previously proposed OptKnock algorithm [87]. Due to their availability, it has become possible to use stoichiometric metabolic models for a variety of tasks, including modeling the organism’s phenotype using methods of flux balance analysis, minimizing metabolic adjustment and its on/off regulation. One of the advantages of OptFlux is the presence of a GUI interface, which considerably simplifies the user’s operation with a mathematical model, unlike COBRA Toolbox and COBRApy, which require programming skills [101].

**MEWpy** is a software package for exploring the different classes of constraint-based models, including metabolic, enzymatic and regulatory models. MEWpy is written in the Python programming language by the developers of Optflux and allows the use of different toolkits, such as GECKO [102] and OptRAM [103], to predict the phenotype of a microorganism and optimize its growth. The advantage of MEWpy is the ability to work with GSM models derived from COBRApy, which simplifies the process of optimization and modification of the mathematical model [95].

**MOST** (Metabolic Optimization and Simulation Tool) is a software written in the Java programming language, which, like Optflux, has a user-friendly GUI interface. Its distinctive feature is the presence of a proprietary GDBB algorithm for searching for gene knockouts to optimize the target product yield, as well as the E-Flux2 and SPOT algorithms [104] to integrate transcriptomic data into a metabolic model with an easy-to-use interface with functions editing like Excel. MOST has a reaction editor with a built-in check for changes to prevent syntax errors when editing reaction equations, as well as the ability to analyze the flux balance, their variability and visualize the resulting calculations on a custom metabolic map represented as a graph of metabolic reactions [89]. The drawback of MOST is the lack of updates and development of the original version of the product.

**In silico discovery** is a commercial software designed for the graphically oriented reconstruction of constrained-based mathematical models, as well as their modification and calculation. The program has a user-friendly interface, an extensive set of tools for model reconstruction by integrating data from different databases and visual control over all integrated reactions, and tools for model tuning and searching for problems associated with the reconstruction, including various cycles, unused (dead-end) metabolites and reactions. There are algorithms for model calculation and optimization, taking into account its kinetic parameters, which makes it possible to expand the constrained-based model into a dynamic one. The main disadvantage of the resource is its unavailability for academic use and, more importantly, its own modeling format, which differs from the widely used SBML (systems biology markup language) [84], which creates the problem of using existing models and independent evaluation of modeling results.

**CAVE** is a web service for integrated calculation, visualization, research and correction of metabolic pathways [91], which can analyze and visualize them for a large number of genome-scale metabolic models using its own graph tool, similar to the analogous tool for visualization in MOST and OptFlux. It has a user-friendly interface that allows editing model responses and the environment for growth when optimizing the model and the cloud server, on which the calculations take place, making it easy to use without the need to install any software or have your own computational capacity. The alternative web application for computation and interactive visualization of fluxes distribution predicted by FBA of GSM models, **Fluxer**, implemented in Python provides different ways for metabolic network representations based on spanning trees, k-shortest paths and complete graphs [90]. Developers of the tool are planning to significantly improve the application capabilities to customize FBA calculations and graph layouts beyond the used methods.

**Cameo** is a constraint-based modeling software package written in the Python programming language [93], based on the previously described COBRApy package, with a slightly different syntax. The package contains integrated OptKnock and OptGene modules, described in the MEWpy library and the OptFlux program, that solves problems of biotechnological engineering, but it lacks evolutionary functions and tasks related to co-optimization functions. It is possible to visualize the model on metabolic maps by integrating the Escherpy library into Cameo, as well as a large set of tools for model analysis and visualization.

**ReFramed** is a constraint-based modeling software package, also written in the Python, which is the refactored version of the previous Frame package. It is based, as in the case of Cameo, on COBRApy and the Escherpy visualization package. Initially, only commercial solvers such as Gurobi and MBI CPLEX (under an academic license) were available for model optimization, but now an Optlang module is available that allows one to connect other solvers of his/her choice. There are tools for model analysis and the ability to extend the model by integrating transcriptomic data with modules for reconstructing context-specific models, such as GIMME and E-Flux (see description below). There is also a module to optimize the analysis of SteadyCom community models [94].

**PySCeS CBMPy** is another constraint-based modeling package written in the Python, which has modules for the classical analysis with FBA and FVA, as well as a multi-threaded FVA variant block. It can significantly accelerate the analysis speed. This package is used as a basis in such programs for reconstructing GSM models as FAME and MetaDraft [96].

**CellNetAnalyzer** is a toolbox written in MATLAB and provides various methods for constraint-based metabolic modeling including metabolic flux analysis (MFA), FBA, flux variability analysis (FVA) and elementary flux modes (EFMs). Moreover, the software package via command line-based operations, or via interactive network maps, provides powerful methods for computational strain design and metabolic engineering [97,98]. This research group also developed several Python-based packages: **CNApy**, a GUI-featured toolbox for metabolic modeling and design of metabolic networks [99], and **StrainDesign**, a single Python platform with a comprehensive set of advanced methods for computational strain design and optimization [100].

Thus, advances in computational methods and continuously developing tools for FBA led to the GSM modeling approach that has become a guiding tool for cell factory design. The diversity of tools and web sources for the analysis of GSM models provides a broad feasibility for development of different metabolic engineering strategies depending on biotechnological requirements and tasks. However, GSM models analyzed in the vast majority of described software are still far from the real metabolic state of cells because the stoichiometric dependencies alone cannot comprehensively reflect the relationships between different metabolic fluxes considering a certain environmental condition that can stimulate a particular regulatory mechanism on transcription or translation levels. To overcome the limitation, a number of methods implemented in computational tools have been proposed that take into account the specificity or context of the cellular state on the different hierarchical levels via the integration of datasets generated by multiomics measurement techniques.

### 2.6. Tools for the Integration of Omics-Data into GSM Models

To date, a large number of programs and algorithms have been developed for the reconstruction of context-specific GSM models. One of the first was the Akesson algorithm [105] developed about 20 years ago. It was based on the deactivation of some reactions in the model according to the level of gene expression. Subsequently, algorithms were developed that employed other methods for reconstructing context-specific models and integrating transcriptomic data into GSM models. Among them are GIMME [13], iMAT [106], MADE [107], INIT [108] and a number of others that have begun to be actively applied for the reconstruction of context-specific GSM models. However, their efficiency was low, comparable to the conventional pFBA optimization algorithm [109]. Thereafter, these algorithms were modified, and more efficient variants of them have emerged, such as GIM^3^E [110], tINIT [111], ftINIT [112] and TIGER [113], new algorithms have evolved the emerging variety of algorithms, such as CADRE [114], CORDA [115], FASTCORE group algorithms [116], deltaFBA [117] and a number of others.

A recent review [14] provides a detailed extended classification of algorithms for reconstructing context-specific metabolic models. According to said review, contemporary algorithms can be divided into four main groups:The GIMME-like group, where most of the methods of this group conduct reconstruction of metabolic models in two steps: the first step is the maximization of a required metabolic functionality (RMF) based on the FBA (or similar) algorithm. The second step is to minimize the penalty function describing the discrepancy between the obtained reaction fluxes and the experimental data while maintaining the flux through the RMF above the given flux fraction. As a rule, the pseudo-reaction of the biomass equation is chosen as the RMF [14]. GIMME-like algorithms include GIMME [13], GIMMEp [118], GIM^3^E [110] and RIPTiDe [119];The iMAT-like family of methods, in contrast to the group above, does not require a precise definition of RMF. This group of algorithms is based on the classification of reactions in the reference model as active or inactive in accordance with the corresponding states in the experimental data, on the basis of which the GSM model is reconstructed. As a consequence, this approach requires that the experimental data be categorized into two or more groups describing different states of the data (e.g., low-expressed and high-expressed in the context of transcriptomics data) [14]. The algorithms of the iMAT-like group include: iMAT [106], INIT [108], ftINIT [112], Lee [120] and RegrEx [121];The MADE-like methods rely on differential expression data in the process of GSM models reconstruction. The last ones describe differences in metabolic fluxes between two contexts/conditions. Similar to the GIMME-like group, the preservation of the minimum flux value required for RMF is also taken into account in these algorithms [14]. Algorithms of the MADE-like group include MADE [107], RMetD2 [122] and deltaFBA [117];The MBA-like algorithms are based on the identification of key reactions and the subsequent removal of reactions that are not part of the core set. Similar to the iMAT-like group, MBA-like algorithms do not have the choice of selecting an RMF, nor do they have the choice of maintaining the flux through it [14]. The MBA-like algorithms include MBA [123] and mCADRE [114], as well as pymCADRE [124], the FASTCORE algorithm group [116] and CORDA [115].

In addition to the algorithms described above, there are a number of others comprising IgemRNA [125], TRFBA [126], E-Flux2 and SPOT [104], PROM [127] and the recently published OVERLAY algorithm [128].

IgemRNA is a library for MATLAB that has a convenient GUI interface and allows to use four methods for reconstructing context-specific models [125]. According to the above classification, IgemRNA is difficult to assign to any group, due to the fact that it uses the approaches of many of the described algorithms. TRFBA and PROM are two similar algorithms that do not require a large amount of data and use transcriptomic data to update reaction boundaries, as well as to enable the identification of regulatory and metabolic networks. These algorithms are based on the method for integrating regulatory and metabolic networks, which are given gene expression data measured under different cultivation conditions. At first, it generates a probabilistic model for constructing a gene regulatory network, which is then integrated into the GSM model by setting flux boundaries proportional to the associated probabilities. One of the shortcomings of these algorithms is that they do not reconstruct the context-specific model, but only give the growth rate after the introduction of constraints and new reaction boundaries [126,127]. According to the classification described above, these algorithms can be classified as GIMME-like. The E-Flux2 and SPOT methods are based on the fact that although the activity of enzymes is not directly determined from the corresponding levels of expression, the latter can be used as an upper bound of the reaction rate. Then, the expression level of each gene is normalizing in the E-Flux2 algorithm by the maximum expression level of the same gene in several experiments. The second algorithm, SPOT maximizes the correlation between a flux vector and its corresponding gene expression data using the Pearson correlation. The authors note that the SPOT algorithm is suitable for cases where the biomass equation is unknown, while the RMF formulation is necessary for the E-Flux2 algorithm [104]. Thus, the E-Flux2 algorithm can also be attributed to the GIMME-like group, and the SPOT algorithm is more similar to the algorithms of the iMAT group. OVERLAY is also a new algorithm based on the use of transcriptomic data for enzyme-constrained models. The authors note the presence of shortcomings of modern algorithms associated with the need to introduce a threshold from the user for expressed genes, as well as the division of genes into highly expressed, medium and low expressed genes. This approach is not suitable for all studied purposes, due to the fact that genes related to the biosynthesis of any target product, e.g., may have a constantly low level of expression. OVERLAY proposes approach that allows one to solve the described problems. Due to the peculiarities of using the enzyme-constrained model, this algorithm should be considered separately from the proposed classification [128].

Thus, the field of algorithms development for the reconstruction of context-specific models is actively evolving. There are different groups of algorithms that have their own advantages and disadvantages when dealing with certain types of data. So, according to [14], GIMME-like algorithms cope better than others in the context of noise robustness for integrating data, and MADE-like algorithms provide the ability to work with differentially expressed genes, which is practically not available in other groups of algorithms. Most of the algorithms for model reconstruction are designed to work with transcriptomic data, but the use of proteomic data is also evolving. Examples of such algorithms are the above-described GIMMEp, iMAT, MAD and tINIT, as well as GECKO3, GECKOpy [129] and OVERLAY [128]. Moreover, not only are algorithms being developed, but also ways of handling them and preparing data for integration. For example, the recently published framework ssGSEAGEM [130] the authors hope to adapt to all existing algorithms for reconstructing context-specific models and thereby unify and simplify the process of reconstruction of such models.

Such an abundance of algorithms for the reconstruction of context-specific models makes it possible to carefully select it for specific tasks. However, it should be noted that most of the algorithms are implemented in the MATLAB language or made to work within its environment, which requires a paid license subscription. In turn, when operating in the Python programming language, which is open source and does not require, in most cases, licensed products, the choice of algorithms is relatively small. They are presented in Table 4.

All the Python algorithms presented above have shown their effectiveness in reconstructing particular context-specific models using omics data, but there is still no algorithm that could use differential gene expression data to reconstruct such models. It should also be noted that the possibility of refining model predictions is limited, despite the availability of programs for automatic reconstruction of GSM models in the Python, libraries for their further modification and optimization, including co-optimization for complex objective functions, which is an important step in tackling biotechnological problems. Furthermore, there are not any context-specific GSM models constructed for methanotrophs to the best of our knowledge. However, a large number of transcriptomic datasets has been accumulated for diverse methanotrophs that are suitable for further integration into previously developed GSM models (see review in [10,52,136,137,138,139,140,141,142,143,144,145,146,147]). Consequently, the application of this advanced approach in constraint-based modeling of methanotrophic metabolism could pave additional ways to novel and potentially transformative solutions for C1-biotechnology in order to improve the competitiveness of methane-consuming bacteria as microbial producers.

The use of Jupyter Notebook [148,149] or its analogues enables to build a full-value pipeline, starting with the processing of transcriptomic data and annotation of the genome, followed by the reconstruction of the constrain-based model at the genome scale level and with the possibility of its extension to a context-specific model. Such unification substantially simplifies the process of handling GSM models. Whereas the presence of Jupyter widgets, for example, makes it possible to implement a simple but convenient user interface, due to which it becomes feasible to check the reproducibility of modeling results. Additionally, it provides the ability to analyze the model for users who do not have programming skills to the proper degree.

## Figures and Tables

**Figure 1 microorganisms-11-02987-f001:**
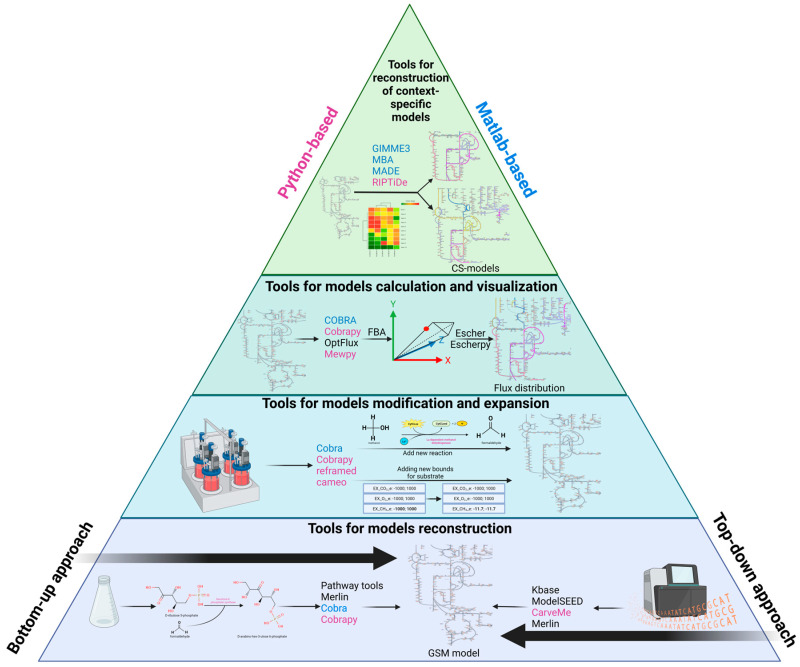
Development stages of a genome-scale metabolic model of any metabolic process (created with BioRender.com). A key step in constraint-based modeling is the construction of the GSM model which is represented at the pyramid’s base. This part of the pyramid briefly illustrates the main approaches (bottom-up: from in vitro data via enzymatic reactions to a metabolic map. Top-down: from omics data to a metabolic map) for GSM models reconstruction. The next block demonstrates an equally important stage—the modification and expansion/reduction of the original GSM model. The block preceding the vertex reflects the model simulation and further visualization of the obtained in silico results using metabolic maps. At the top of the pyramid is a relatively new stage that provides a significant refinement of the model’s predictions through the integration of omics data into the original GSM model for the reconstruction of context-specific models (CS models). Tools developed using the Python programming language are highlighted in pink, while software packages written in MATLAB are highlighted in blue.

**Table 1 microorganisms-11-02987-t001:** GSM models constructed to investigate metabolism of C1-utilizing bacteria. Models in the table are given in chronological order.

Organism	ID *	Genes	Reactions	Metabolites	Tools & Databases **	References	Memote Score ***	Cobrapy Model Consistency ^#^
*Methylobacterium extorquens* AM1		–	67	65		[48]		
	*i*RP911	911	1139	977	CellNet Analyser MicroScope MetaCyc KEGG	[49]	-	-
*Methylotuvimicrobium buryatense* 5G	*i*Mb5G (B1)	–	841	–	Pathway-Tools MicroScope	[50]		
	*i*Mb5GB1 update	314	402	403	COBRA Toolbox	[51]	42%	98%
*Methylotuvimicrobium alcaliphilum* 20Z^R^	*i*IA409	409	436	423	COBRA toolbox KEGG, BiGG, BioCyc	[24,52]	25%	94.5%
*Methylococcus capsulatus*	*i*McBath	730	898	877	Cobrapy KEGG, BiGG MetaCyc	[53]	54%	66.3%
	*i*MC535	535	899	865	ModelSEED COBRA Toolbox KEGG, MetaCyc	[26]	21%	60.1%
*Methylocystis hirsuta* CSC1		2478	1399	1460	ModelSEED Cobrapy KEGG	[54]	19%	50.75%
*Methylocystis sp. SC2*		2251	1449	1434	ModelSEED Cobrapy KEGG	[54]	19%	49.82%
*Methylocystis sp. SB2*		2281	1380	1453	ModelSEED Cobrapy KEGG	[54]	19%	50.86%
*Methylocystis parvus OBBP*		2795	1326	1399	ModelSEED Cobrapy	[55]	19%	53.1%
*Methylosinus trichosporium OB3b*	*iMsOB3b*	683	1043	1020	Cobrapy KEGG	[56]	23%	67.24%
*Methylocella silvestris BL2*		681	1436	1474	ModelSEED Cobrapy	[57]	19%	48%
*Methylomicrobium album BG8*	*iJV806*	803	1358	1367	KBase COBRA Toolbox Cobrapy KEGG CycleFreeFlux [58]	[59]	27%	53.52%

* Column *ID* contains the standard identifier of the reconstructed GSM model if it has one. ** Column Tools & Databases comprises a list of software and sources used for the model reconstruction and analysis. *** Column Memote score represents the total score which is calculated by the Memote tool and describes a model quality assessment based on the number of independent tests. The metrics of the individual tests are added up to a weighted sum of all test results normalized by the maximally achievable score equal to 100%. ^#^ Column Cobrapy model consistency reflects the percentage of unblocked reaction in the model, where 100% means it is a completely consistent model.

**Table 3 microorganisms-11-02987-t003:** Tools for GSM model modification and optimization.

Program	Tool Type	Algorithms for Optimization	Programs Availability	Reference
COBRA Toolbox 3.0 https://github.com/opencobra/cobratoolbox	Matlab package	FBA, pFBA, dFBA, dynamic rFBA, geometricFBA, relaxed FBA, FVA, MOMA, ROOM, FASTCORE, thermo FBA, looples FBA	Available, but needed Matlab	[86]
OptFlux http://www.optflux.org	GUI based program	FBA, pFBA, FVA, MOMA, LMOMA, ROOM, MiMBL, OptRAM, OptGene, OptKnock.	Available	[87]
MOST http://most.ccib.rutgers.edu/	GUI based program	FBA, FVA, E-Flux2, SPOT	Available, but last update 5 years ago	[88,89]
In silico discovery https://www.insilico-biotechnology.com/	GUI based program	FBA, FVA	Commercial	
Fluxer https://fluxer.umbc.edu/	Web-service	FBA	Available	[90]
CAVE https://cave.biodesign.ac.cn/	Web-service	FBA, FVA	Available	[91]
Cobrapy http://opencobra.sourceforge.net/	Python package	FBA, pFBA, dFBA, geometric FBA, relaxed FBA, FVA, MOMA, ROOM, FASTCORE, thermodynamic FBA, looples FBA	Available	[92]
cameo http://cameo.bio. http://try.cameo.bio	Python package	FBA, FVA, OptKnock, OptGene	Available	[93]
ReFramed https://github.com/cdanielmachado/reframed	Python package	FBA, FVA, pFBA, FBrAtio, CAFBA, MOMA, lMOMA, ROOM, looples FBA, thermodynamic FBA, TVA, NET, GIMME, E-Flux, SteadyCom	Available	[94]
Mewpy https://github.com/BioSystemsUM/mewpy	Python package	FBA, pFBA, FVA, MOMA, LMOMA, ROOM, MiMBL, OptRAM, OptGene, OptKnock	Available	[95]
PySCeS CBMPy https://cbmpy.sourceforge.net/	Python package	FBA, FVA	Available	[96]
CellNetAnalyzer (CNA) https://www2.mpi-magdeburg.mpg.de/projects/cna/cna.html	MATLAB toolbox	MFA, FBA, FVA, EFM, Yield analysis, Strain optimization (CASOP)	Available	[97,98]
CNApy https://github.com/cnapy-org/CNApy	Python package	FBA, pFBA, FVA, EFM, Yield optimization, Computational strain design (OptKnock, RobustKnock, OptCouple and advanced Minimal Cut Sets), OptMDFpathway, thermodynamic FBA, phase plane analysis	Available	[99]
StrainDesign https://github.com/klamt-lab/straindesign	Python package	FBA, pFBA, FVA, OptKnock, RobustKnock, OptCouple, general minimal cut set (MCS) approach, cRegMCS, FOCAL, ModCell2	Available	[100]

**Table 4 microorganisms-11-02987-t004:** Tools for the reconstruction of context-specific GSM models implemented in the Python.

Program	Data Type	Requirements	Examples of Use
RIPTiDe https://github.com/mjenior/riptide	Transcriptomic	GSM model, transcriptomics data file	[119,131]
pymCADRE https://github.com/draeger-lab/pymCADRE/	Transcriptomic Metabolomic	GSM model, list of precursor metabolites, confidence scores, list of gene IDs for all genes in model, list of ubiquity scores calculated for all genes in model	[124,132]
Troppo https://github.com/BioSystemsUM/troppo	Transcriptomic	GSM or enzyme-constrained model, multi-omics datasets	[133,134]
Geckopy3.0 https://doi.org/10.1101/2023.03.20.533446	Proteomic	Enzyme-constrained model, kinetics and omics data	[129]
A new GIMME–Based method	Transcriptomic	GSM model, transcriptomics data file	[135]

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
