# Peer review of "State-of the-Art Constraint-Based Modeling of Microbial Metabolism: From Basics to Context-Specific Models with a Focus on Methanotrophs"

_microorganisms, 2023, doi:10.3390/microorganisms11122987_

Round 1
Reviewer 1 Report
Comments and Suggestions for Authors
Kulyashov et al. offered a thorough examination of methodologies and tools employed in the metabolic modelling of methanotrophic bacteria. The authors discuss the general principles of metabolic modeling and the available resources (databases, software, and packages) used for reconstruction of metabolic models. Review also highlights the variety of metabolic models developed for methanotrophic bacteria. The manuscript is well-structured and offers a good introduction to the field of metabolic modelling with a reference to methane metabolism.
However, there are several points where the manuscript could be improved.
- The authors may consider to mention Genbank database in the section “Databases of the microorganisms’ genomes”
- Line 122: it is more than 20,000 PGDBs according to Biocyc webpage
- Lines 129-130: rephrase the following “that summarize information on genes, pathways of their regulation”
- Line 173: rephrase the following “and the information provide in them serve”
- Line 178: replace “diverse species and strains” with “diverse strains”
- Line 179: replace “through” with “using”
- Line 182: same as line 178. The models were built for specific strains.
- Lines 191-193: Consider rephrasing
- Line 204: “to test”
- Lines 222-227: Consider rephrasing
- Lines 228-229: It is unclear what is meant by the term “alternative” when referring to Methylomicrobium album BG8
- Line 237: replace “state-of the-art” by ”state-of-the-art”
- Lines 237-238: consider rephrasing, the term “Memote score” may be used
- Lines 243-246: revise the sentence
- Line 251: a few words about methanotrophic archaea could be added
- Line 272: consider rephrasing “a connected graph, and with Escher metabolic maps “
- Line 281: “the Gap-filling algorithm, like Kbase”. This part appears ambiguous as Kbase is not an algorithm.
- Line 286: “epy” ?
- Lines 280-290: consider rephrasing
- Lines 310-313: consider rephrasing
- Lines 388-392: revise this sentence
- Lines 395-396: revise this sentence
- Lines 519-523: revise this part. The text structure is not consistent
- Line 527: consider rephrasing
- Lines 537-539 : consider rephrasing
- Lines 556-560: consider dividing this part into at least two or more sentences.
- Lines 566-568: revise this sentence
- Lines 574-579: revise this sentence
Comments on the Quality of English Language
Some sentences would benefit from minor English editing
Author Response
The file with replies to reviewers is attached.

Reviewer 2 Report
Comments and Suggestions for Authors
The study entitled “State-of the-art in constraint-based modeling of microbial metabolism: from basics to context-specific models with a focus on methanotrophs” provided an overview of various computational strategies implemented for microbial systems, and summarized computational tools and web-resources that can streamline the constraint-based microbial metabolic modeling from the reconstruction of GSM models to the development of context-specific metabolic models considering transcriptomic datasets to fine-tune the flux bounds of an original model to a specific experimental context. Overall, the study is novel and timely, and it will serve as a pointer to other similar studies in different regions of the world in future. However, the authors have listed and introduced a large number of relevant software and websites, which is indeed a good thing for some readers who would like to carry out related work, but with such a large number of software, websites and methods, what are the "specific" strengths and weaknesses of different methods and software in conducting research on this type of microorganisms, in the case of "methane-trophic bacteria" proposed by the authors? What are the "specific" advantages and disadvantages of the different methods and software in carrying out research on this type of microorganisms, and please illustrate these issues with specific analyses and data. Since the authors also mention that the genome-scale mathematical modeling of metabolism has been envisioned as one of the most productive strategies for the integration of muti-scale data to better understand methane metabolism and enable its biotechnological implementation. So, please elaborate using methanotrophic bacteria.
Author Response

(The authors gave the same response as above.)

Reviewer 3 Report
Comments and Suggestions for Authors
A review article titled ‘’State-of-the art in constraint-based modeling of microbial metabolism: from basics to context- specific models with a focus on methanotrophs’’ by Kulyashova M.A. Kolmykova S.K., Khlebodarova T.M. and Akberdin I.R. was based on the analysis of 141 scientific publications, mainly from the last 5 years. If enzymatic metabolic modeling can be considered one of the most productive strategies in biotechnology, then the proposed methodology should be used step by step and improved. The publication presents an overview of various computational strategies for methanotrophic systems. The work contains a short introduction and chapters describing the reconstruction strategies and analysis of the metabolic model at the genome scale. Numerous databases and online portal resources have been analysed and are used for the initial reconstruction of metabolic pathways, as well as a summary of each resource.
I am confident that the data in this review article may be helpful to researchers involved in model microbial metabolism.
I believe that the review article can be published in the form presented.
Author Response

(The authors gave the same response as above.)

Reviewer 4 Report
Comments and Suggestions for Authors
The manuscript “State-of the-art in constraint-based modeling of microbial metabolism: from basics to context-specific models with a focus on methanotrophs” by MA Kulyashov, SK Kolmykov, TM Khlebodarova, and IR Akberdin summarizes the existing body of research, and available tools and methods for the reconstruction of genome-scale models, mostly of microorganisms, and for analysis thereof. The application of this approach is illustrated exemplary for methanotrophs.
The manuscript is mostly comprehensive and logically structured with some exceptions (see comments below). While in general sufficiently precise, the descriptions are not always specific (also see comments below), which in some cases may prevent their immediate understandability.
Early in the manuscript the term systems biology is used synonymously with synthetic biology, however, they are not the same. A clearer differentiation and more specific use of the terms should be implemented to avoid confusion.
There are some problems with English language, most of which I believe I have caught and pointed out in my comments, which follow below.
Line 12: the greenhouse gas
Line 17: waste = trash/rubbish; what you mean is vast
Line 31: I’m unsure what de-novo synthesis pertains to here – are you suggesting the bottom-up construction of a living organism? This has not been accomplished so far.
Line 39: “etc.” is NOT an acceptable expression in scientific literature. Specify what you want to say or omit.
Line 42: I believe synthetic biology would be the more appropriate term here.
Line 42: conversion of it
Line 46: not sure what “bounds” is supposed to mean here
Line 64: information is a plural word à are
Figure 1: it should be “expansion” on the second level.
Line 114: NCBI / GenBank should probably be part of 2.2. (Databases of microorganisms’ genomes). For metabolic functions also UniProt is often very useful.
Line 120: an
Line 177 – 181: recommend breaking up this rather lengthy sentence
Line 191: unclear what “The last ones” pertains to
Line 192: I think it should be “which” instead of “that”
Line 199: some non-trivial categories like ID, Memote Score, Cobrapy model consistency, should be explained in the caption of the table.
Line 206: …a fatty acid producer …
Line 214: it seems sort of arbitrary here to mention that the medium contained Ca or La – how is this important in the current context? If it is not of further significance, I’d suggest to omit the statement.
Line 219: further
Line 220: how is the name of the PI and University of relevance here? Especially since none of the other laboratories have ben mentioned by name this seems oddly uncommon and inconsistent.
Line 201 – 233: The different GSM studies in these paragraphs (and table 1) seem to be described in no particular order. Maybe start with type I methanotrophs, i.e. MC Bath, then the Methylomicrobia, followed by the type II methanotrophs and methylotrophs. Or do it the other way round. Another way could be to describe them in chronological order, from the first study to the most recent one. If that’s what has been done a statement in the caption of table 1 in that regard could be meaningful.
Line 237: It should be briefly described what the MEMOTE score is and how to interpret it – without that the analysis is not going to be very useful to most readers.
Line 244: exchange of what? sink of what?
Line 247: what is FROG analysis and how is it informative of the quality of a metabolic model?
Line 250-254: this should be with the description of the other methanotrophs’ models above
Line 308: existed
Line 303: It features
Line 386: It is unclear what the authors mean by “most demanded resources” – if it is to give an indication as to which tools are most widely used, some evidence for that should be given, e.g. by the number of citations of the main publication associated with the tool.
Line 466: the
Line 476: the
Line 482: Elementary Flux Mode Analysis (EMA)
Line 604: the
Line 551: 4 = four
Comments on the Quality of English Language
There are some problems with English language, most of which I believe I have caught and pointed out in my comments.
Author Response

(The authors gave the same response as above.)
